Genome comparison implies the role of Wsm2 in membrane trafficking and protein degradation

Zhang Guorong 1
Hua Zhihua hua@ohio.edu 2 3
1 Agricultural Research Center-Hays, Kansas State University , Hays , KS , United States of America
2 Department of Environmental and Plant Biology, Ohio University , Athens , OH , United States of America
3 Interdisciplinary Program in Molecular and Cellular Biology, Ohio University , Athens , OH , United States of America
Uversky Vladimir
Electronic publication date: 2018 Apr 23
Publication date: 2018
Volume: 6
Electronic Location ID: e4678
Received 2018 Mar 9; Accepted 2018 Apr 9
Copyright: ©2018 Zhang and Hua
Copyright year: 2018
Copyright holder: Zhang and Hua
License: This is an open access article distributed under the terms of the Creative Commons Attribution License, which permits unrestricted use, distribution, reproduction and adaptation in any medium and for any purpose provided that it is properly attributed. For attribution, the original author(s), title, publication source (PeerJ) and either DOI or URL of the article must be cited.
License URL: https://creativecommons.org/licenses/by/4.0/

Keywords: Triticum aestivum, Oryza sativa, Brachypodium distachyon, Wheat streak mosaic virus, Wsm2, Orthologous groups, Bioinformatics, Intrachromosomal recombination

Funding: USDA National Institute of Food and Agriculture competitive grant 2017-67007-25939 Hatch grant 1001453 Ohio University Start-Up grant SU1007172 This work was supported by the USDA National Institute of Food and Agriculture competitive grant (2017-67007-25939 to Guorong Zhang) and Hatch grant (1001453 to Guorong Zhang), and Ohio University Start-Up grant (SU1007172 to Zhihua Hua). The funders had no role in study design, data collection and analysis, decision to publish, or preparation of the manuscript.

==============================
Wheat streak mosaic virus (WSMV) causes streak mosaic disease in wheat (Triticum aestivum L.) and has been an important constraint limiting wheat production in many regions around the world. Wsm2 is the only resistance gene discovered in wheat genome and has been located in a short genomic region of its chromosome 3B. However, the sequence nature and the biological function of Wsm2 remain unknown due to the difficulty of genetic manipulation in wheat. In this study, we tested WSMV infectivity among wheat and its two closely related grass species, rice (Oryza sativa) and Brachypodium distachyon. Based on the phenotypic result and previous genomic studies, we developed a novel bioinformatics pipeline for interpreting a potential biological function of Wsm2 and its ancestor locus in wheat. In the WSMV resistance tests, we found that rice has a WMSV resistance gene while Brachypodium does not, which allowed us to hypothesize the presence of a Wsm2 ortholog in rice. Our OrthoMCL analysis of protein coding genes on wheat chromosome 3B and its syntenic chromosomes in rice and Brachypodium discovered 4,035 OrthoMCL groups as preliminary candidates of Wsm2 orthologs. Given that Wsm2 is likely duplicated through an intrachromosomal illegitimate recombination and that Wsm2 is dominant, we inferred that this new WSMV-resistance gene acquired an activation domain, lost an inhibition domain, or gained high expression compared to its ancestor locus. Through comparison, we identified that 67, 16, and 10 out of 4,035 OrthoMCL orthologous groups contain a rice member with 25% shorter or longer in length, or 10 fold more expression, respectively, than those from wheat and Brachypodium. Taken together, we predicted a total of 93 good candidates for a Wsm2 ancestor locus. All of these 93 candidates are not tightly linked with Wsm2, indicative of the role of illegitimate recombination in the birth of Wsm2. Further sequence analysis suggests that the protein products of Wsm2 may combat WSMV disease through a molecular mechanism involving protein degradation and/or membrane trafficking. The 93 putative Wsm2 ancestor loci discovered in this study could serve as good candidates for future genetic isolation of the true Wsm2 locus.

Introduction

Wheat streak mosaic virus (WSMV) causes streak mosaic disease in wheat (Triticum aestivum L.) and has been reported in many regions around the world (Fahim et al., 2011; Sharp et al., 2002). The WSMV is transmitted by wheat curl mites (WCM: Aceria tosichella Keifer) (Navia et al., 2013) and wheat is the preferred host for both WCM and WSMV (Baleya et al., 2001; Murray & Brennan, 2009). WSMV-infected wheat plants develop yellow leaf streaks during early infection and the symptom could spread to the entire leaf if the virus is not effectively controlled. Stunted growth is also common in severely infected plants. Price et al. (2010) found that WSMV infection could reduce root development and affect water use efficiency. WSMV is one important constraint limiting wheat production in the Great Plains of United States. The average yield loss in this region was estimated about 2% per year (Appel et al., 2013; Christian & Willis, 1993). However, up to 13% reduction in wheat production due to WSMV disease has been reported in Kansas, USA (Sim, Willis & Eversmeyer, 1988). In severe cases wheat production could be completely destroyed by WSMV (McNeil et al., 1996).

Unfortunately, effective chemicals are not yet available for controlling WSMV and its WCM vector (Tan et al., 2017). Host resistance is the primary and effective way to suppress WSMV. To date, three WSMV resistance genes, Wsm1, Wsm2, and Wsm3, have been identified. Both Wsm1 and Wsm3 were found in a wild relative, Thinopyrum intermedium (Host) Barkworth & D.R. Dewey, and they have been introduced into the wheat genome through translocation (Gill et al., 1995; Triebe et al., 1991). However, alien translocation often results in yield penalty due to the incorporation of non-adapted genes. For example, lines introgressed with Wsm1 showed various yield reductions ranging from 11 to 28% (Sharp et al., 2002), limiting the breeding application of this type of resistant sources. Wsm2 was discovered in a wheat breeding line CO960293-2 (Haley et al., 2002). Genetic studies have shown that the WSMV resistance in CO960293-2 is controlled by a single dominant allele (Wsm2) and it has been genetically mapped on chromosome arm 3BS of the wheat genome (Lu et al., 2011). Recently, our group and others have further located Wsm2 into 0.4 cM region flanked by BS00022387_51 and BS00088683_51 using a dense microarray containing 90,000 single nucleotide polymorphic (SNP) sites (Assanga et al., 2017). However, its sequence nature and biological function still remain elusive.

Wsm2 has been introduced into several wheat cultivars to acquire WSMV resistance without compromising yield, such as “RonL” (Martin et al., 2007), “Snowmass” (Haley et al., 2011), “Clara CL” (Martin et al., 2014), “Oakley CL” (Zhang et al., 2015), and “Joe” (Zhang et al., 2016a), demonstrating a great potential in improving WSMV resistance. Through allelic test and Wsm2-linked marker analysis in nine wheat lines we have shown that Wsm2 and/or its tightly linked genes are primarily responsible for the WSMV resistance (Zhang, Seifers & Martin, 2015; Zhang et al., 2016b). Interestingly, the functions of three WSMV-resistant genes, Wsm1, Wsm2, and Wsm3, are all temperature sensitive, which hold their proper WSMV resistant functions up to 20, 18, and 24 °C, respectively (Gill et al., 2008; Seifers et al., 2013a; Seifers et al., 2013b; Seifers et al., 1995). A recent study indicated that the replication and movement of WSMV, and the disease symptom development were greatly affected by temperature (Wosula et al., 2017). It remains unclear whether the temperature-dependent effectiveness of WSMV resistance genes is due to the variable pathogenesis of WSMV under different temperature conditions and/or caused by the temperature-mediated gene expression and functional regulation. Only if we isolate these genes would we be able to address these questions more closely. In this work, we applied cross species phenotypic and genomic analyses and discovered that Wsm2 might encode a protein involved in membrane trafficking and protein degradation.

Materials and Methods

WSMV infectivity tests on rice and Brachypodium distachyon

Rice (Oryza sativa ssp. Japonica) cultivar Nipponbare, B. distachyon (Brachypodium hereafter) accession Bd21-3, WSMV-resistant wheat cultivar RonL, and WSMV-susceptible wheat cultivar “Tomahawk” were seeded in rows in two metal flats (21 × 31 cm) filled with a potting mix (Sungro, Vacouver, Canada). Each line had two replications with 12 seeds per replication (row) in each flat. At the two-leaf stage, plants were mechanically inoculated (finger-rubbing) with a WSMV isolate, Sidney 81. Inoculum preparation was done as described in Seifers et al. (2006). In brief, infected wheat leaf tissues were grounded at a 1:10 (wt/vol) dilution in 0.02 M potassium phosphate buffer (pH 7) and filtered through cheesecloth. This extract was used as inoculum after adding abrasive (Crystolon flour B, 600 mesh; Norton Co., Worcester, MA, USA) with a concentration of 0.01 g/mL. This method of inoculum preparation and inoculation was used throughout the study. After inoculation, two flats were maintained in different growth chambers (Percival Model PGC-15WC) with one set at 18 °C and the other kept at 22 °C under a short-day photoperiod condition (12 h fluorescent light (250 µEs−1m−2) and 12 h darkness). Four weeks after inoculation, indirect enzyme-linked immunosorbent assays (ELISA) were conducted for each plant as described in Seifers et al. (2006). Two leaf tissue bulks from non-inoculated Tomahawk plants (healthy check) were included as healthy controls. The GHV value (Sample ELISA value / Healthy ELISA value) was calculated for each plant. The plant was considered as susceptible if its GHV was greater than 2 (Seifers et al., 2006). The percentage of resistant plants was calculated for each line in each replication. The ANOVA was conducted for the percentage of resistant plants using GLM model with SAS 9.4 and the least significant difference (LSD) at α = 0.01 was used to conduct comparison among four genotypes.

A further infectiveness test was conducted on additional 44 Brachypodium accessions (Gordon et al., 2017) to examine if there is any variation of WSMV resistance. All 44 accessions plus Bd21-3 were planted in rows in two metal flats (30 × 50 cm) with eight seeds per row. Each accession was planted in one row while an additional row of Bd21-3 was planted as the healthy check. At the two-leaf stage, all plants except for the healthy check were mechanically inoculated with Sidney 81 as described above. After inoculation, both flats were maintained in a growth chamber set at 18 °C with the same short-day photoperiod as aforementioned. Four weeks after inoculation, ELISA tests were conducted for each plant including the healthy check plants. The GHV was calculated and used to determine the WSMV susceptibility or resistance. The percentage of susceptible plants (infection rate) was calculated for each accession.

Back assay with inoculum made from infected Bd21-3

Symptomatic leaf tissues from Bd21-3 were bulked and six different dilution rates of inoculums (1:5, 1:10, 1:20, 1:40, 1:80, and 1:160 wt/vol) were made as described above. A new batch of Bd21-3 seedlings were raised at 18 °C under a short-day growth condition. Bd21-3 plants were grown in seven rows in one metal flat (21 × 31 cm). At the two-leaf stage, each row was inoculated with a different dilution. The last row was not inoculated and used as the healthy check. Four weeks after inoculation, ELISA tests were conducted on every plant. The GHV of each inoculated plant was calculated and used to determine the WSMV susceptibility or resistance. The percentage of susceptible plants (infection rate) was calculated for each dilution rate.

Orthology relationship analysis

To define orthologous genes in Brachypodium, rice, and wheat that likely encode Wsm2, the protein sequences of previously annotated genes on Brachypodium chromosome 2, rice chromosome 1, and wheat chromosome 3B were retrieved from each genome project (Choulet et al., 2014; International Brachypodium Initiative, 2010; Kawahara et al., 2013). The sequences were combined and analyzed for orthology relationships based on their similarities using OrthoMCL (Li, Stoeckert Jr & Roos, 2003). Briefly, an all-against-all BLASTp search (Altschul et al., 1990) was performed to find sequence similarity between each pair of sequences. The resulting sequence similarity matrix was subjected to a Markov Cluster Algorithm (MCL) clustering analysis to define orthologous groups among three species. An inflation value of 1.5 was identified to be the best to yield all putative OrthoMCL groups among three species analyzed.

Expression analysis

We used the number of expression sequence tags (EST) to represent the relative expression level of an orthologous gene in Brachypodium, rice, and wheat. To identify the number of ESTs of each orthologous member, the EST sequences of each species were downloaded from the EST database at Genbank (https://www.ncbi.nlm.nih.gov/nucest). The coding sequence of an orthologous gene from each species was used as a query for BLASTn search against its EST database (Altschul et al., 1990). An EST was considered to reflect a true expression of a query gene if (1) it had >95% identity to the query coding sequence, (2) the aligned sequences cover at least 75% of the EST or the query sequence, and (3) at least 50 nucleotides of the EST was included in the alignment (Hua et al., 2011). To compare expression of orthologous genes across three species, the absolute EST value of each gene was normalized by the total EST number of the corresponding species.

Functional prediction of a putative Wsm2 candidate

The protein sequence of a putative Wsm2 candidate gene was used as a query to search against the Pfam-A protein-protein interaction database (https://pfam.xfam.org, Version 31) by HMMER3, an accelerated profile hidden Markov model (profile HMM) search tool (Eddy, 2011). The presence of a predicted Pfam-A protein-protein interaction domain (e-value cutoff ≤ 1) was used to categorize the putative biological function of a candidate.

Identification of the physical position of a Wsm2 ancestor locus

The physical position (coordinate) of a putative Wsm2 ancestor locus was retrieved based on the Generic Feature Format (GFF3) file from the wheat chromosome 3B genome project (Choulet et al., 2014). The distribution of putative Wsm2 ancestor loci were visualized by plotting each locus on chromosome 3B.

Results

Absence of WSMV resistance genes in wild species

To examine whether a WSMV-resistant gene could be generated through spontaneous natural mutations in a wild species, we asked whether Brachypodium, a strictly self-pollinated species, is resistant to WSMV. Since all three up-to-date identified Wsm loci are temperature sensitive, we carried out the WSMV infectiveness tests on Brachypodium at two different temperatures, 18 °C and 22 °C, at which the Wsm2-containing wheat cultivar RonL shows WSMV resistant and susceptible phenotypes, respectively (Seifers et al., 2013b).

In the initial screen (Table 1), the susceptible wheat check cultivar Tomahawk was severely infected at both temperature conditions, indicating the effectiveness of the WSMV isolate Sidney 81 in this study. As a positive control, over 90% of the RonL plants were resistant to WSMV at 18 °C while all individuals displayed a susceptible symptom at 22 °C, confirming the temperature-sensitive phenotype of Wsm2 in RonL. Among 35 Brachypododium Bd21-3 plants examined (17 plants at 18 °C and 18 plants at 22 °C), 80% (14 plants at each temperature) were susceptible to WSMV based on ELISA tests (GHV > 2, which indicates WSMV susceptibility (Seifers et al., 2006)). Statistically, no significant difference (p < 0.01, ANOVA test) was observed between WSMV-susceptible Tomahawk and Bd21-3 at both temperatures (Table 1), suggesting that Bd21-3 does not express any WSMV resistance gene. To confirm the susceptibility of Bd21-3 to WSMV, we performed a back assay, which used extracts from infected Bd21-3 plants as pathogen sources. To further understand the dynamic infection of WSMV, the original extract was diluted in a series of concentrations and used to inoculate Bd21-3 seedlings. The overall infection rates ranged from 50 to 100% (Table 2). The first five dilutions were very infective and have infected most of inoculated plants (83.3∼100%). Therefore, our data indicate that Bd21-3 is WSMV susceptible.

Table 1 Resistant percentages (%) of rice cultivar Nipponbare, Brachypodium accession Bd21-3, wheat cultivars RonL and Tomahawk after inoculation with WSMV isolate Sidney 81.

Temperature	Genotype	Rep Ia	Rep II	Meanb	
18 °C	Nipponbare (rice)	83.3 (12)	100.0 (8)	91.7A	
	Bd21-3 (Brachypodium)	20.0 (10)	14.3 (7)	17.2B	
	RonL (Wheat)	90.0 (10)	90.9 (11)	90.5A	
	Tomahawk (wheat)	0.0 (11)	0.0 (11)	0.0B	
	LSD (0.01)			39.7	
22 °C	Nipponbare (rice)	100.0 (2)	100.0 (1)	100.0A	
	Bd21-3 (Brachypodium)	12.5 (8)	30.0 (10)	21.3B	
	RonL (wheat)	0.0 (12)	0.0 (11)	0.0B	
	Tomahawk (wheat)	0.0 (10)	0.0 (10)	0.0B	
	LSD (0.01)			36.1	
Notes.

a Number in the parenthesis indicates the size of sample in each replication.

b Genotypes not having the same letter in common are significantly different at p < 0.01.

Table 2 Back assay of WSMV infection on Bd21-3 with a series of inoculum dilutions.

Inoculum dilution rate (wt/vol)	Total plants	Infected plants	Infection rate (%)	ELISAa	GHVb	
1:5	9	8	88.9	0.29 ± 0.11	26.1 ± 10.4	
1:10	8	8	100.0	0.33 ± 0.06	29.7 ± 5.2	
1:20	8	8	100.0	0.29 ± 0.04	26.4 ± 3.6	
1:40	6	5	83.3	0.24 ± 0.13	22.0 ± 12.0	
1:80	7	7	100.0	0.27 ± 0.05	24.4 ± 5.0	
1:160	8	4	50.0	0.13  + 0.13	11.6  + 11.9	
Healthy control	5			0.01 ± 0.01	1.0 ± 0.53	
Notes.

a ELISA, enzyme linked immunosorbent assay, mean absorbance ± SD.

b GHV, Sample ELISA value/Healthy ELISA value, mean ± SD.

Since new sources of temperature-sensitive resistance to WSMV have been identified from a large collection of wheat accessions (Seifers et al., 2013a), we then asked whether any Brachypodium natural variants could possess a WSMV-resistant gene. In total, 44 Brachypodium accessions with extensive genetic variations (Gordon et al., 2017) were selected for WSMV-infectiveness analysis (Table 3). All accessions together with Bd21-3 were grown under the same condition (see Materials and Methods) and inoculated with the same WSMV isolate Sidney 81 under 18 °C as described above. Four weeks after inoculation, ELISA tests were conducted to examine the WSMV-susceptibility or -resistance of individuals. Unexpectedly, all the accessions had a 100% infection rate and all of them displayed a greater ELISA value than the uninoculated Bd21-3 control and large GHVs ranging from 56.3 to 112.1 (Table 2), suggesting that Brachypodium might not contain any genetic sources for WSMV resistance and that a WSMV resistance gene is not likely attributed to spontaneous natural mutations.

Table 3 Infectivity variation of WSMV on 45 Brachypodium natural variants.

Accessions	Total plants	Infected plants	Infection rate (%)	ELISAa	GHVb	
ABRS	7	7	100	0.54 ± 0.09	78.9 ± 12.6	
ABR4	7	7	100	0.55 ± 0.05	80.4 ± 7.5	
Adi-2	4	4	100	0.60 ± 0.10	88.6 ± 14.0	
Adi-10	5	5	100	0.49 ± 0.07	71.6 ± 10.6	
ARN1	9	9	100	0.53 ± 0.08	78.0 ± 11.5	
Bd1-1	5	5	100	0.57 ± 0.05	83.6 ± 6.9	
Bd2-3	5	5	100	0.50 ± 0.06	73.1 ± 9.2	
Bd3-1	5	5	100	0.54 ± 0.05	79.1 ± 7.4	
Bd21-1	7	7	100	0.52 ± 0.23	76.8 ± 33.9	
Bd21-3	7	7	100	0.59 ± 0.08	87.2 ± 11.5	
Bd29-1	9	9	100	0.40 ± 0.17	58.9 ± 24.3	
Bd30-1	7	7	100	0.50 ± 0.03	73.2 ± 4.7	
Bis1	5	5	100	0.58 ± 0.04	85.5 ± 6.4	
Foz1	3	3	100	0.50 ± 0.03	74.1 ± 3.7	
Gaz8	5	5	100	0.51 ± 0.04	75.6 ± 6.3	
Kah-1	6	6	100	0.48 ± 0.03	71.0 ± 4.2	
Kah-S	6	6	100	0.47 ± 0.08	69.2 ± 11.6	
Koz1	8	8	100	0.48 ± 0.06	70.8 ± 8.5	
Koz3	7	7	100	0.50 ± 0.11	73.4 ± 15.6	
Luc1	7	7	100	0.48 ± 0.10	71.2 ± 14.7	
Mig3	9	9	100	0.49 ± 0.04	71.3 ± 6.5	
Mon3	5	5	100	0.76 ± 0.12	112.1 ± 17.8	
Mur1	7	7	100	0.43 ± 0.21	63.5 ± 30.2	
Per1	7	7	100	0.53 ± 0.05	77.8 ± 7.0	
RON2	8	8	100	0.61 ± 0.06	89.6 ± 8.5	
Sig2	7	7	100	0.60 ± 0.08	87.8 ± 11.7	
TEK-2	7	7	100	0.45 ± 0.05	66.4 ± 6.6	
TEK-4	8	8	100	0.64 ± 0.07	94.3 ± 9.8	
TEK11	7	7	100	0.59 ± 0.07	86.2 ± 10.7	
TR2B	7	7	100	0.55 ± 0.03	81.2 ± 4.9	
TR3C	8	8	100	0.61 ± 0.04	89.9 ± 6.5	
TR7a	5	5	100	0.42 ± 0.24	62.4 ± 34.6	
TR8i	9	9	100	0.46 ± 0.17	66.9 ± 24.5	
TR9K	7	7	100	0.44 ± 0.28	65.2 ± 41.3	
TR10C	6	6	100	0.60 ± 0.08	87.6 ± 11.4	
TR11A	8	8	100	0.55 ± 0.07	80.4 ± 10.5	
TR11G	8	8	100	0.51 ± 0.05	74.8 ± 7.7	
TR12C	5	5	100	0.38 ± 0.20	56.4 ± 29.6	
TRBa	6	6	100	0.58 ± 0.12	85.5 ± 17.4	
TR13C	8	8	100	0.63 ± 0.10	91.2 ± 14.9	
TR26	6	6	100	0.53 ± 0.08	77.9 ± 12.3	
TRIi	9	9	100	0.58 ± 0.05	85.8 ± 7.5	
TRSi	7	7	100	0.61 ± 0.05	90.1 ± 8.0	
Uni2	8	8	100	0.68 ± 0.12	100.5 ± 17.7	
18-1	6	6	100	0.57 ± 0.04	83.1 ± 6.3	
Healthy control	5	0	0	0.0068 ± 0.008		
Notes.

a ELISA, enzyme linked immunosorbent assay, mean absorbance ± SD.

b GHV, Sample ELISA value/Healthy ELISA value, mean ± SD.

Presence of a WSMV resistance gene in rice

Since Brachypodium is a naturally self-pollinated wild species (International Brachypodium Initiative, 2010) and wheat is a crop, we next asked whether another crop species, rice (Oryza sativa), could contain a WSMV resistance gene. Both wheat and rice have been domesticated for over 10,000 years (Meyer, DuVal & Jensen, 2012) and it is known that domestication has significantly changed genome arrangement of crops from their wild relatives by fixing elite agronomic traits that benefits agricultural production (Chantret et al., 2005). We chose Nipponbare as a test rice cultivar because of the availability of its well-annotated genome (Kawahara et al., 2013). We performed WSMV infectiveness tests on Nipponbare together with Brachypodium and wheat lines at both 18 and 22 °C (see Materials and Methods). Interestingly, among 20 Nipponbare seedlings examined at 18 °C, all of them had a similar resistant percentage as RonL (Table 1), indicating the presence of a WSMV-resistant gene in rice genome. Surprisingly, all rice plants remained resistance to WSMV at 22 °C while RonL did not. Thus, rice might have a different resistance gene or allele than Wsm2.

Identifying candidates of Wsm2 orthologs in Brachypodium, rice and wheat

The missing of an effective WSMV resistant gene in 45 natural populations of Brachypodium suggests that WSMV resistant genes are not likely generated through single nucleotide polymorphic or short insertion/deletion mutations, which often arise from random natural mutations. Comparative genomic analysis has revealed high inter- and intrachromosomal gene duplication rates in the wheat genome, particularly in chromosome 3B (Choulet et al., 2014; Dubcovsky & Dvorak, 2007). This high recombination rate might contribute to the birth of a WSMV resistance gene, especially Wsm2, which was gained through a three-way cross hybridization of susceptible parental lines CO850034, PI222668, and TAM107 (Seifers et al., 2006). It is likely that exon shuffling through DNA recombination in the process of breeding gave rise to a new function of an ancestor Wsm2 locus for WSMV resistance. The discovery of WSMV resistance in Nipponbare inbreed line and the syntenic relationship between rice chromosome 1 and wheat chromosome 3B implied that Nipponbare might encode a Wsm2 homologous, which was gained through exon shuffling on chromosome 1 during the breeding process. Given the dominant function of Wsm2 allele (Lu et al., 2011), exon shuffling resulted in the ancestor Wsm2 locus to (1) lose an ancestral inhibition domain, (2) acquire an activation domain, or (3) increase expression. Since Wsm2 was produced only through four generations of segregation, intrachromosomal recombination is more likely to happen than interchromosomal recombination to give the birth of Wsm2 because the frequency of DNA paring between two separate chromosomes is lower than that within a chromosome.

The large genome size of wheat has limited its genetic manipulation. In order to isolate Wsm2, we developed a bioinformatics pipeline to predict the candidates of a Wsm2 ancestor locus and its orthologs in Brachypodium and rice (Fig. 1). Since wheat chromosome 3B is syntenic to Brachypodium chromosome 2 and rice chromosome 1, amino acid sequences of protein coding genes annotated on these three chromosomes were retrieved from each genome project. In total, 5,070, 7,074, and 7,264 protein sequences were obtained from Brachypodium (Bd21-3) (International Brachypodium Initiative, 2010), rice (Nipponbare) (Kawahara et al., 2013), and wheat 3B (Choulet et al., 2014) genomes, respectively. These sequences were then combined for an OrthoMCL analysis (Li, Stoeckert Jr & Roos, 2003) and 4,035 OrthoMCL groups were resolved as preliminary candidates of Wsm2 orthologs (Fig. 1, File S1). Surprisingly, we did not find any potential rice orthologs of Wsm2 described in a previous study (Tan et al., 2017), although our list did include all Brachypodium genes from the same work, indicating that the previous orthology analysis could be problematic.

Figure 1 A diagram showing the analysis procedures and summary of results in identifying Wsm2 ancestor candidates.

Prediction of a Wsm2 ancestor locus

Wsm2 ancestor locus might acquire a new function for WSMV resistance through deletion of a repression domain, acquisition of an activation domain, or upregulation of expression (Fig. 1). Therefore, we reasoned that a putative Wsm2 ortholog in rice would be 25% shorter or longer in length, or 10 fold higher in expression than its orthologs in Brachypodium and wheat. Based on these criteria, we first compared the protein sequence length differences between a rice ortholog and the other members from wheat and Brachypodium within the same orthologous group. In total, we found that 67 and 16 out of 4,035 OrthoMCL orthologous groups contain a rice sequence that are 25% shorter or longer, respectively, than those from wheat and Brachypodium (Table S1). Therefore, the wheat members from these two groups represent good candidates of a Wsm2 ancestor locus.

The gain of WSMV resistant function in Wsm2 could be also attributed to a gene dosage-dependent response. One simple way to increase Wsm2 transcripts is through fusion of an ancestor Wsm2 to a strong promoter via recombination. To identify the possibility of this process, we counted the number of ESTs for each member in each of 4,035 OrthoMCL orthologous groups (File S1). If rice expresses a Wsm2 ortholog responsible for its WSMV resistance, which is gene dosage-dependent, we reason that the expression of this ortholog would have a significant higher expression than its orthologs in wheat and Brachypodium. To find these orthologs, we compared their expression across three species based on normalized EST values (see Materials and Methods). In total, 17,661, 37,590, and 69,162 ESTs were retrieved from the genomes of Brachypodium, rice, and wheat, respectively. Through BLASTn search (Altschul et al., 1990), we identified 10 rice genes that have 10 fold more normalized ESTs than their corresponding orthologous members in Brachypodium and wheat (Table S1). Taken together, we predicted a total of 93 good candidates for a Wsm2 ancestor locus.

The Wsm2 ancestor locus is not likely linked with Wsm2

To identify the linkage relationship of a Wsm2 ancestor locus with Wsm2, we retrieved the physical positions (coordinates) of all 93 candidates and plotted them on chromosome 3B (Fig. 2). As a control, the positions of eight SNP markers tightly linked with Wsm2 were also identified (Assanga et al., 2017) (File S2). Although none of our candidate genes are tightly linked with Wsm2, we identified 10 genes that reside in the R1 and R3 distal regions (Fig. 2), two regions with high recombination rates on wheat chromosome 3B (Choulet et al., 2014). With respect of the overall high recombination rate of chromosome 3B, we cannot rule out the possibility of the remaining 83 candidates to be a Wsm2 ancestor locus. Certainly, the closer to the centromere region a candidate gene is, the less likely it could be a Wsm2 ancestor locus.

Figure 2 Physical relationship of 93 candidates of Wsm2 ancestor loci with Wsm2 locus on wheat chromosome 3B.

(A) Positions of 16 candidates that may acquire an activation domain to become Wsm2. (B) Locations of 67 candidates that may form Wsm2 by deletion of an inhibition domain. (C) Distribution of 10 candidates that may be changed as Wsm2 through fusion with a strong promoter in the Wsm2 locus. (D) The place of the Wsm2 locus on chromosome 3B that is represented by its tightly linked 8 SNP markers. (E) Schematic representation of the structure of chromosome 3B adopted from Choulet et al. (Choulet et al., 2014). Circle dots: 93 candidates of Wsm2 ancestor loci. Black diamonds: eight SNP markers that are tightly linked with Wsm2. Red diamonds, beginning and end of chromosome 3B; R1, R2, and R3, three regions of chromosome 3B with different recombination rates; C, centromeric/pericentromeric region; Red/yellow shaded regions, two distal regions (R1 and R3) with high recombination rates. ±, Watson/Crick DNA strands of chromosome 3B.

Putative function of a Wsm2 candidate gene

Since all 93 candidates could be a Wsm2 ancestor locus, we further analyzed the functional domains in each protein sequence using HMMER3 (http://hmmer.org) to search against Pfam-A protein-protein interaction database (https://pfam.xfam.org, Version 31). Based on the broad function of each domain, we classified the putative functions of 93 candidates into seven categories, including glycosylation and membrane trafficking, protein ubiquitylation and degradation, transcription factor, chaperone, exonuclease, epigenetic regulation, and unknown (Table S2). Among these, we found that 11 and 17 candidates likely play a role in protein degradation (including ubiquitylation) and membrane trafficking (including glycosylation), which are 5.5 and 8.5 fold more than the third large known functional category (transcription factor), respectively. Therefore, proteins encoded by Wsm2 may combat WSMV disease through a molecular mechanism involving protein degradation and/or membrane trafficking.

Discussion

Conditions that may influence the result of WSMV infectivity test

In the WSMV infection tests, it is easy to determine the viral infectivity in wheat through visible disease symptoms (streaks or mosaic) on leaves. However, such symptoms are not easily observed in Brachypodium because of its small-sized leaves. Therefore, we used the ELISA tests to assist evaluation. In the initial testing, few small Brachypodium plants (1∼3 plants in each replication) were not infected due to inoculation challenges on narrow and skinny leaves. In the later test of 45 Brachypodium accessions, fertilizer was applied to stimulate robust and healthy plant growth, which allowed effective finger-rubbing inoculation on leaves. Not surprisingly, 100% infection rate was detected on all 45 accessions, including Bd21-3 that was used in the initial test. Therefore, healthy and large leaf area is important for evaluating WSMV infectivity in grass species.

Seed purity or temperature fluctuations could also impact the infectivity result due to temperature sensitivity of Wsm2. For example, in the initial testing, RonL did not show 100% resistance to WSMV at 18 °C (Table 1), which could be explained by the problems of either seed purity or temperature fluctuations of the growth condition. This is not uncommon in WSMV-infectivity test. A similar result was observed for Wsm2-containing wheat line CO960293 in previous studies (Lu et al., 2011; Seifers et al., 2013b). In addition, weak plants may cause WSMV infectivity/growth. For example, in the course of our WSMV infectivity tests, we detected GHV greater than 2 (2.1 and 6.5, Table 1, Rep I) in two small rice plants, which indicates WSMV susceptibility (Seifers et al., 2006). However, in other replications, all rice plants examined remained healthy and displayed 100% resistance to WSMV (Table 1). Collectively, our data suggest that replicates and number of individuals are important to give a comprehensive evaluation of WSMV infectivity tests.

The contribution of crop domestication in the birth of Wsm2

Brachypodium is evolutionarily close to wheat. However, the former is present naturally and is strictly self-pollinated wild species (Asplund, Hagenblad & Leino, 2000; International Brachypodium Initiative, 2010) and the later is a crop species that has been domesticated for ∼12,000 years (Asplund, Hagenblad & Leino, 2000; Meyer, DuVal & Jensen, 2012). The domestication process has significantly increased genome rearrangement and produced contrastive genome architecture of a domestic species comparing to its wild-type progenitor genome (Yue et al., 2017). Both the lack and the presence of a WSMV resistance gene in Brachypodium and rice, respectively, suggest that a WSMV-resistant trait is more likely a product of crop domestication. Consistent with this hypothesis, various wild grass species have been shown to be susceptible to WSMV due to the lack of genetic sources (Ito et al., 2012).

Wsm2 is not likely linked with its ancestor locus

Previous functional prediction suggested that a Wsm2 ancestor locus is linked to Wsm2 (Tan et al., 2017). However, our broad orthologous group analysis did not identify any rice genes described in Tan et al. (2017) that share an ortholog in wheat (File S1). In addition, all Brachypodium genes discovered in their work have either a similar length or a comparable expression level as the orthologous members in wheat (Table S1), further lowering the possibility of close genetic linkage between Wsm2 and its ancestor locus. Since our cross species analysis of WSMV resistance suggested that Wsm2 likely gained the pathogen resistant function through intrachromosomal recombination (Table 1 to Table 3), we inferred that a Wsm2 ancestor is not necessarily linked with Wsm2.

Through comparative genome analysis, a previous study has discovered that the wheat Hardness (Ha) locus was a rejoining product of DNA fragments separated from two different loci via illegitimate recombination (Chantret et al., 2005). Such recombination events could occur in any genomic region, which is not necessary related to transposon-mediate DNA insertion/deletion (Gregory, 2004; Kirik, Salomon & Puchta, 2000). The identification of wide distribution of short conserved sequence motifs at rearrangement breakpoints suggested that illegitimate recombination between unlinked genomic regions is a major evolutionary driving force in wheat domestication (Chantret et al., 2005). Therefore, the unlinkage of our 93 candidates of Wsm2 ancestor locus with Wsm2 indicates that the Wms2 ancestor locus is not necessary within the Wsm2 locus and that the birth of Wms2 is likely attributed to illegitimate recombination on chromosome 3B.

The role of protein degradation and membrane trafficking in pathogen defense

The discovery of many Wsm2 ancestor candidates expressing a domain involved in protein degradation and membrane trafficking is intriguing. Recent genetic, genomic, and proteomic studies have highlighted the role of these two biochemical mechanisms in plant pathogen defense at various stages, including perception, response, and defense (Duplan & Rivas, 2014; Furlan, Klinkenberg & Trujillo, 2012; Furniss & Spoel, 2015; Li, Lu & Shan, 2014; Marino, Peeters & Rivas, 2012). Through genome annotation, we have discovered that the ubiquitin-26S proteasome system (UPS) is extremely large in the wheat genome, in part due to its polyploidy nature (unpublished result). For example, we discovered that wheat genome encodes the largest family of ubiquitin and ubiquitin-like genes in 50 plant genomes (Z Hua, P Doroodian & W Vu, 2018, unpublished data), further indicating the importance of the UPS in regulating wheat development and growth. In addition, the role of protein ubiquitylation is also intimately connected with membrane trafficking in cells (Clague & Urbe, 2017). Therefore, our study implies a putative role of Wsm2 in ubiquitylation and/or membrane trafficking-mediated protein degradation.

Conclusions

In this study, we compared WSMV resistance among three closely related grass species (Tables 1–3) and developed a novel bioinformatics pipeline for predicting potential candidates of a Wsm2 ancestor locus (Fig. 1). Given that Wsm2 is likely duplicated through an intrachromosomal illegitimate recombination and the dominant phenotype of Wsm2, we inferred that this new WSMV-resistant gene acquired an activation domain, lost an inhibition domain, or gained elevated expression compared to its ancestor locus (Table S1). The resulting 93 putative Wsm2 ancestor loci could serve as good candidates for future genetic isolation of the true Wsm2 locus. We may design new polymerase chain reaction (PCR) primers based on the nucleotide sequences of each candidate to examine the presence of an additional copy that is linked to the Wsm2 locus. The finding of such a copy could serve a starting point to clone the full-length of a putative Wsm2 gene via thermal asymmetric interlaced PCR (Liu et al., 1995). This may provide an efficient way to isolate and characterize the molecular function of Wsm2.

Supplemental Information

File S1 List of 4,035 orthologous groups encoded in Brachypodium chromosome 2, rice chromosome 1, and wheat Chromosome 3B

Click here for additional data file.

File S2 Nucleotide sequences of 8 SNP markers tightly linked with Wsm2

Click here for additional data file.

Table S1 Sequence and expression comparison of 93 Wsm2 ancestor candidates with Brachypodium and rice orthologous members

Click here for additional data file.

Table S2 Putative biochemical functions of 93 Wsm2 ancestor candidates

Click here for additional data file.

Table S3 Experimental raw data for WSMV infectivity tests on Nipponbare and Bd21-3

Click here for additional data file.

Table S4 Experimental raw data for WSMV infectivity tests on 45 Brachypodium natural variants

Click here for additional data file.

Table S5 Experimental raw data for back assay on Bd21-3

Click here for additional data file.

This is contribution number 18-373-J from the Kansas Agricultural Experiment Station. We thank Daniel P. Woods (University of Wisconsin-Madison) for providing the seeds of 45 Brachypodium accessions and Jeff Ackerman (Kansas State University) for valuable assistance in virus testing.

Additional Information and Declarations

Competing Interests

Author Contributions

Data Availability

The authors declare there are no competing interests.

Guorong Zhang conceived and designed the experiments, performed the experiments, analyzed the data, contributed reagents/materials/analysis tools, prepared figures and/or tables, authored or reviewed drafts of the paper.

Zhihua Hua conceived and designed the experiments, performed the experiments, analyzed the data, contributed reagents/materials/analysis tools, prepared figures and/or tables, authored or reviewed drafts of the paper, approved the final draft.

The following information was supplied regarding data availability:

The raw data are provided in Tables S1–S5.

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
