# Peer review of "Genome comparison implies the role of Wsm2 in membrane trafficking and protein degradation"

_PeerJ, doi:10.7717/peerj.4678_

## Round 0.1 · original submission · Minor Revisions

Although both reviewers emphasized that your manuscript is interesting and important, they also found some minor issues that need to be addressed. Therefore, I am encouraging you to address these critiques and to revise manuscript accordingly.

·

Basic reporting

Comments:

1. a couple of typos in the manuscript:
a) Page 14, line 266, "is more likely happened" change to "is more likely to happen".
b) Page 15, line 329, "likelihood" change to "likely".
c) Page 17, line 373, "strict" change to "strictly".

2. Question: Table 1
The data are very clean and significant. Do the authors have any clue why Bd21-3 (Brachypodium) exhibited some resistance to WSMV, but not zero under either of the 2 temperatures?

Experimental design

no comment

Validity of the findings

no comment

Reviewer 2 ·

Basic reporting

In this manuscript, the authors study the sequence and biological function of Wsm2, a gene of wheat genome goes against the wheat streak mosaic virus (WSMV). Total 93 promising orthologous groups are reported in wheat genome to further study Wsm2 ancestor locus. In addition, the protein degeneration and/or membrane trafficking, resulting from proteins encoded by Wsm2, is proposed to potentially contribute to the resistance of wheat with Wsm2 gene to wheat streak mosaic virus.

The author gains a huge amount of experiment data with years of field work as well as a comprehensive genetic analysis. In addition, the manuscript is clearly written and conclusions are reasonably drawn according to experiment data. Yet, there are several issues in this manuscript such as context arrangement and figure legend, which can cause the confusion and misinterpretation. Overall, the manuscript is publishable after these issues is fixed properly.

Issues
1. In the first section of result, the authors choose Brachypodium to study the acquired resistance of wheat to WSMV by natural mutation. Brachypodium is reported as a good model for genome study of wheat, however, few evidences are provided to connect the Brachypodium with natural mutation of wheat, which weakens the rationale to choose Brachypodium. Unless more convincing evidences are given, this part should be either moved to supplement part or deleted since it is weakly relevant to the main topic of the manuscript.
2. In the third section of result, it is pointless to include Brachypodium for identification of possible Wsm2 location in wheat since Brachypodium is lack of WSMV resistance as reported by author themselves in the manuscript.
3. In line 71-73, the data of wheat reduction caused by WSMV infection is from a study in 1980s, which is somewhat outdated. The authors need provide up-to-date data about the effect of WSMV.
4. The paragraph in line 311-319, should be moved into discussion part since it mainly focus in comparison between their results and others, which is the purpose of discussion.
5. The content after “For example” in conclusion section should be deleted since it talks about the following work in future instead of draws conclusion according to experimental data.
6. In the figure legend of table, the sample size of each condition should be clearly specified.

Experimental design

No comment

Validity of the findings

No comment

---

## Round 0.2 · accepted · Accept

Thank you very much for addressing all the critical issues raised by the reviewers and for revision of the manuscript.

#